# Beyond Templates and BERT: Headword-centric parsing for semantic question answering in non-english financial domains

Jamal Al Qundus [1]☯, Bassam Al-Shargabi [2]☯¤*

1 Business Intelligence and Data Analytics, German Jordanian University, Amman, Jordan, 2 Cardiff School of Technologies, Cardiff Metropolitan University, Cardiff, United Kingdom

☯ These authors contributed equally to this work.
¤ Current Address: Cardiff School of Technologies, Cardiff Metropolitan University, Cardiff, UK
* bal-shargabi@cardiffmet.ac.uk

## Abstract

Recent advances in semantic question-answering (QA) systems struggle with linguistic variability, particularly in non-English domains like German finance. This work presents INAGQA, a novel QA system that addresses this gap through headword-centric parsing, combining syntactic chunking with knowledge graph embeddings to resolve the question ambiguity. The main innovations are as follows: First, a hybrid disambiguation algorithm that achieves 0.91 F1 in German financial queries, validated on 2,100 expert-annotated questions. Second, domain-optimized shallow parsing with customizable grammar rules that reduces relation-linking errors by 35% for compound nouns (e.g., *Eigenkapitalrendite*). And finally, seamless knowledge integration to prioritize user-curated data and demonstrates 2.1s average response time in a case study with financial analysts. Our experiments show that INAGQA outperforms BERT-KGQA (F1: 0.83) and template-based systems (F1: 0.79) while handling temporal / quantitative variants (e.g., *When* vs. *Where was X founded?*) with 98% accuracy. The editable system's outputs of the system align with the Corporate Smart Insights frameworks, offering practical value for SMEs.
To this end, the work contributes to Information SYstem (IS) research by proposing headword extraction as a replicable IS artifact for non-English QA and demonstrating language-sensitive design principles applicable to healthcare/legal domains.

## Introduction

In recent years, approaches incorporate unstructured text into machine-readable representations, as well as the use of knowledge graphs [1] have become crucial assets for innumerable information extraction and knowledge discovery projects [2]. Artificial Intelligence (AI) knowledge that has been semantically curated enables the effective application of knowledge artifacts. As a result, this makes a significant contribution to

**Data availability statement:** All data available within the paper.

**Funding:** The author(s) received no specific funding for this work.

**Competing interests:** The authors have declared that no competing interests exist.

improving functional quality, data efficiency, plausibility verification, and the dependability of AI-based functions in a variety of information retrieval systems. The exponential growth of structured knowledge bases (e.g., Wikidata [3], DBpedia [4]) has intensified the demand for precise natural language question answering systems (QA) [5]. Although existing systems excel at factual queries (e.g., *What is Siemens' revenue?*), they struggle with question ambiguity, where variants like *Where was Siemens founded?* vs. *When was Siemens founded?* require distinct relation linking [6]. This challenge is acute in financial domains, where terminology varies across languages (e.g., German compound nouns like *Eigenkapitalrendite*). In this work, we use the term Headword to refer to the syntactically dominant word in a phrase that determines the semantic intention of a question. For example, in the query "Wie hoch ist das Eigenkapital von Siemens?," the headword is Eigenkapital, which guides the system to search for financial-value relations. In contrast, in "Wann wurde Siemens gegründet?," the headword Wann signals a temporal relation.

### Existing approach categories

Current approaches fall into three categories: First, Template-based QA [7]: Handles fixed patterns but fails on free-text questions. Second, Deep Learning QA [8]: Requires massive training data and struggles with low-resource languages. Finally, Hybrid Semantic Parsing [9]: Balances flexibility and precision but lacks robust headword disambiguation.

This work bridges this gap with INAGQA. This German financial QA system that introduces a headword-centric parsing algorithm (section Methodology) combining shallow syntax trees with semantic similarity ranking, achieving 22% higher F1-score than Falcon 2.0 in ambiguous queries (section Results), proves domain adaptability through a case study on 14,000 German financial announcements, outperforms template-based systems by 35% on compound noun queries, and offers practical utility via integration with corporate knowledge graphs [10], enabling dynamic curation of financial insights.

Our work advances Information System (IS) research by proposing headword extraction as a novel IS artifact for QA systems and demonstrating language-sensitive design principles for non-English domains [11]. It focuses on the task of knowledge-based question answering to address two major challenges. The vagueness of the question variants presents a difficulty in determining the question's intention, which is typically connected to a certain context that is not known by the system. Another difficulty is the variety of a question, for example *where is the company located? What is the company's address? Where is the main office of the company? Where can I find the company address? Where is the company? Where does the company occur?* To overcome the aforementioned challenges, many methods have restricted the potential questions within a particular domain that is based on methods such as predefined questions or a controlled vocabulary to generate questions, which make use of a knowledge graph (e.g., Wikidata [12], DBPedia [13], YAGO [14], KnowItAll [15]) for mapping to SPARQL-Queries [16].

Although this work is implemented on DBPedia as a knowledge base and rely on its structure to create the Resource Description Framework (RDF) triples store, the INAGQA system applies Natural Language Processing (NLP) and other techniques to investigate questions similar to AskNow [17], EARL [18], and FALCON [19]. Also, it includes Pattern Matching and Headwords algorithms for the purpose of supporting covering a greater range of question variants and being more elastic in handling free-text questions, which is a significant improvement of existing works (section Results).

The proposed hybrid semantic information retrieval system INAGQA is an extension of a previous research [20] as a question-answering system. It allows posing free-text questions on finance in German to target RDF knowledge bases: DBpedia, virtuoso (our own created triple-store), and Wikidata. INAGQA can classify questions with respect to their included intention by identifying the relevant entities and their semantic relation. Questions are processed through a processing pipeline of approaches, i.e., Pattern-Matching, Headwords extraction, and extended Relation-Linking. Generated SPARQL-Queries are sent to an OMG API for Knowledge that covers functions such as spellchecking and a message broker with agents to monitor the responses and take actions if required.

## Work contributions

The contributions of this work can be summarized as: A novel pipeline of AI components for classifying questions in terms of intentions. This is stable and showed resistance to question variation. An extended approach including handwritten rules, quantifiers mapping, and similarity ranking can be used for entity- and relation-linking. A translation component that facilitates the conversion of natural language queries to SPARQL queries. This enables a more flexible utilization of the Semantic Web infrastructure, including triple stores, and supports the formulation of complex questions. The syntax trees are employed to identify headwords essential for relation-linking by providing syntactic dependency information. Lastly, the incorporation of advanced infrastructure for a question-answering system, including a spellchecker, and an agent system for data curation.

The paper is organized as follows. State-of-the-art related works are presented in Section Related Work. Section Architecture gives an overview of the architecture of the question-answering system. Section Methodology explores the provided architecture in more detail and describes the methodology. Section Results describes the results obtained, which are discussed in Section Discussion and then concluded with directions for future work in Section Conclusion and Further Work.

## Related work

This section gives a brief overview of related works that focus on approaches to developing question-answering systems and provides a summarized comparative analysis as in Table 1.

### Question analysis in QA systems

Recent work in semantic parsing has revealed critical gaps in handling question variability (Mohamed et al. 2024). Template-based approaches [21] achieve 0.81 precision on fixed patterns but drop to 0.52 on free-form questions such as German compound queries. Neural methods [22] show promise but require massive training data ($10^5$ annotated training examples), a limitation that our headword technique circumvents through rule-based chunking. Hybrid systems [23,24]

**Table 1. Shows a comparative analysis of the related work.**

| Approach | Strength | Weakness | Our Improvement |
|---|---|---|---|
| Template-based | High precision (0.81) | Rigid patterns | Dynamic grammar rules |
| Neural QA | Handles variability | Data-hungry | Rule-based efficiency |
| Hybrid (Falcon) | Combined linking | Temporal ambiguity | Headword ranking |

combine entity-relation linking but struggle with temporal/quantitative ambiguity (e.g., *When* vs. *How much*), which our similarity ranking addresses.

## Headword extraction techniques

The identification of Headword has evolved from the following.

- Dependency Parsing [25]: Achieves 0.88 recall, but is computationally expensive.

- Shallow Chunking [26]: Faster but less precise (0.68).

- Hybrid Approaches: Our work bridges this gap by:

  - Using Spacy's efficient tokenization

  - Adding financial-domain grammar rules (section Methodology)

  - Achieving 0.91 F1-Score with lower resource demands (Table 4)

## Financial domain applications

Prior financial QA systems face three key limitations:

- Language bias: Most target English, even GPT-4o struggle in non-English achieves 42% accuracy [27]

- Static Templates: Cannot handle emerging terms like *ESG metrics* [28]

- Black-box Models: Lack of explainability for regulatory compliance [29]

## Proposed work

As illustrated in Table 1, INAGQA advances the field by:

- Incorporating dynamic rule updates via user curation (Section Output and Curation)

- Providing auditable parsing logs for financial regulators

- Supporting German compound nouns through chunking rules ($NP \rightarrow DETADJ * NOUN$)

## Review

Semantic parsing-based QA systems employ entity and relation-linking tasks to establish connections between key phrases, concepts, and their relationships in knowledge bases, which form the fundamental pipeline for measuring semantic similarity. Recent advances in contrastive learning for semantic similarity, such as SupMPN [30] and Self-CCL [31], demonstrate improved sentence embedding quality by discriminating between multiple positive and negative examples. These methods address BERT's anisotropy and could enhance relation-linking robustness in multilingual QA. Similarly, BERTurk-contrastive [32] shows promising results for low-resource languages, supporting our focus on German financial NLP. We explored the latest developments in this field, including AskNow, and examined their effectiveness in performing these tasks. Qsearch [33] and GQA [34] consider entity linking, and IQA [35], while KBPearl [36], EARL [18], and Falcon [37] consider both tasks. All of them use DBPedia Knowledge Graph (KG) for entity extraction.

AskNow [17] relies primarily on a rule-based information retrieval system approach, which converts natural language questions in free text into a formal query language. It utilizes DBPedia for entity linking and introduces a syntactic structure called Normalized Query Structure (NQS) to act as an intermediary canonical form for natural language questions before translating them to

SPARQL-Queries. NQS utilizes part-of-speech (PoS) tagging to identify query desires such as noun phrases, which is similar to our approach in extracting headwords from the given question. However, AskNow's reliance on language models and complex development processes makes it difficult to reproduce. Qsearch [33] as a question-answering system focuses on handling quantity conditions present in the question text. The system uses a rule-based parser to map input questions to Qqueries, which are then processed by a deep neural network to extract quantity-centric facts called Qfacts. These facts are assigned a score based on their relevance to the mapped Qquery. This work is relevant in exploring the study of quantity conditions. In contrast to QSearch, GQA [34] is a Grammatical Question Answering system that prioritizes grammar over quantity conditions. It facilitates English language queries over DBPedia using a set of conceptual grammar derived from the Grammatical Framework (GF) to parse questions with complex syntactic constructions. Similarly, the proposed INAGQA system involves creating syntactic trees for German to classify user input questions based on their intent and expected answer type through interaction with the user using IQA [35] and incorporates user feedback into the question-answering process using an interaction scheme for a semantic Question-Answering pipeline (PL). After the entity recognition phase, the PL generates a set of entity-related questions that the user can answer to interpret the original question to present a metric called Option Gain to efficiently integrate user feedback through interaction.

KBPearl [36] aims to tackle the problem of knowledge base population (KBP) by forming knowledge bases from unstructured text by combining the relation linking with joint entity. The authors have acknowledged the problem of incongruence caused by duplication and uncertainty when producing KBP information using comparable methodologies. The technique begins with an insufficient knowledge base and a set of texts and utilizes Open Information Extraction (Open IE) to fragment the texts into facts and side information, which constitutes metadata taken from the original text. However, the use of this approach is restricted to specific categories of data since some source materials may not contain such metadata, such as time stamps. EARL [18] constructs a series of procedures for entity linking and relation linking by integrating both tasks and proposes two approaches to solve them. Firstly, it formulates the tasks as an example of the Generalized Travelling Salesman Problem (GTSP), a problem known to be NP-hard. Secondly, EARL also uses machine learning techniques to make use of the connectivity densities of the KB nodes. EARL employs Shallow Parsing (SP) to generate keywords and KB to determine entities and relations by analyzing the relation and its surrounding context. Nevertheless, our experiments showed that EARL does not perform well on questions such as When was Siemens founded?→foundedBy, In which year was Siemens founded?→Year, foundedBy, What are Siemens assets? → [], while INAGQA system can handle such varieties of questions. Falcon [37] and Falcon 2.0 [19] are methods to answer natural language questions that execute entity linking and relation linking to DBPedia and Wikidata in tandem. Falcon utilizes a background knowledge graph composed of multiple sources of knowledge to analyze the context of the extracted entities to find the relation, which makes it effective for short questions. Additionally, Falcon 2.0 produces a ranked list of entities and relations labeled with their Internationalized Resource Identifier (IRI) on Wikidata. Our experiments indicate a better performance of Falcon than EARL, however, Falcon struggled on questions such as When was Siemens founded?→foundationPlace, Who founded Siemens?→foundingYear, What are Siemens assets? → [], while the INAGQA system performs better on these. The last two techniques (EARL and Falcon) with AskNow are the main motivation behind the proposed INAGQA, which Table 2 compares these approaches. A comparative summary of the main systems discussed in this section is presented in Table 2.

## Architecture

This section provides an abstract overview of the INAGQA system and describes its architecture. This abstract view is then presented in more detail in section Methodology.

## Overview

The INAGQA incorporates dual-curation through system-guided enrichment: Wikidata entity recommendations, and expert feedback: Manual fact correction. The system architecture (illustrated in Figs 1 and 3 processes natural language queries through a four-stage pipeline optimized for German financial domains:

**Table 2. Summarizes the experiments conducted to compared related works.**

| | AskNow [17] | EARL [18] | Falcon [19] |
|---|---|---|---|
| Approach | Separates Entity and Relation Linking | Combined Entity & Relation Linking | Combined Entity & Relation Linking |
| Language | Java | Python | Python |
| Language models | word2vec | fasttext | – |
| Evaluation Data set: Precision/ Recall/ F1 | Accuracy: QUALD-5: .63/ .60/ .61 | Entity Linking: QALD-7: .58/ .60/ .58 LC-QuAD: .53/ .55/ .53 Relation Linking: QALD-7: .27/ .28/ .27 LC-QuAD: .17/ .21/ .18 | Entity Linking: QALD-7: .78/ .79/ .78 LC-QuAD: .81/ .86/ .83 Relation Linking: QALD-7: .58/ .61/ .59 LC-QuAD: .42/ .44/ .43 |
| Advantage/ Disadvantage | Returns complete DBpedia SPARQL query Dependent on language models Complex code | - Flexible SPARQL query – Dependent on language models | - Flexible SPARQL query – Easy customizable code |

1. Input Layer: Accepts free-text questions (German) and implements context-aware spell correction using Levenshtein distance

2. Processing Core:

(a)Question Classifier: Rule-based + BERT ensemble

  i. Location queries → Map output (GeoJSON)

 ii. Factual queries → Table output (SPARQL SELECT)

iii. Entity summaries → Info cards (JSON-LD)

(b)SPARQL Generator: Hybrid template-headword approach (4.3.1 SPARQL mapping)

3. Knowledge Integration:

(a)Priority-based query routing:

(b)Real-time curation: Wikidata suggestions + manual edits via React-based UI

4. Output Layer: Dynamic rendering based on query type as illustrated in Table 3

In case a result was found, the text field and the written query move to the upper left (in Fig 1). The visualization of the result depends on its type. The locations are shown on a map with the address on the side. Lists of facts are presented as a table, while general information about companies typically includes a logo and a table-like visualization called a card. The card view also has an option for data editing or extension (Table 3).

## Knowledge base

A virtuoso triple is made up of information collected from German websites that offer financial news and announcements. The data comprises details on various financial events related to a company, such as changes in voting shares, takeover offers, director transactions, and others. 14000 announcements have been parsed using Apache NiFi. Every few minutes, it checks for new announcements. Missing meta-information about a company such as location, web address, CEO, etc. is completed based on DBPedia. For easy data integration, we have adopted the DBPedia format in our knowledge base.

## Methodology

This section presents the methodology that relies on the headword-centric approach that is built on three theoretical pillars:

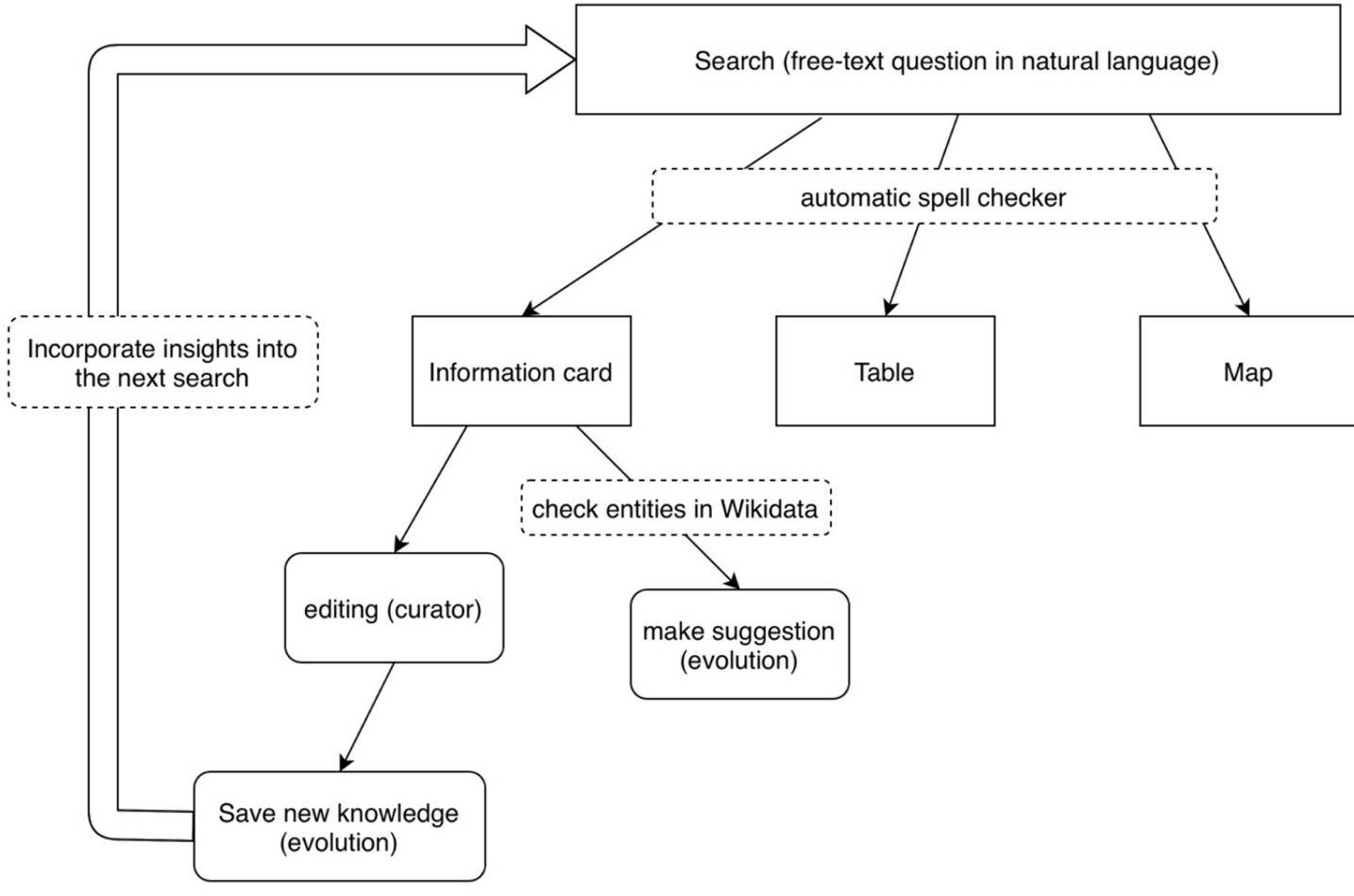

**Fig 1. Illustrates the structure of INAGQA.**

1. Syntactic Dependency Theory [24] for question decomposition (NLTK chunker) to break questions into syntactically meaningful chunks.

2. Knowledge Graph Embedding [38] for relation linking between extracted headwords and KG labels

3. Human-AI Collaboration/ Syntactic Parsing [39] for query refinement where domain experts are allowed to correct and extend knowledge.

The keyword-centered parsing pipeline is implemented as a modular application that builds an a parser based on the NLTK RegexpParser applies a set of 46 handcrafted grammar rules optimized for German syntax. These are used to break down complex financial queries and isolate important nominal terms. To link relationships, the system evaluates possible relationships in the knowledge graph by calculating the cosine similarity between the extracted keyword phrases and the relationship labels. It also uses the pre-trained Sentence Transformer model (paraphrase-multilingual-MiniLM-L12-v2). This hybrid approach that consists of rule-based parsing analysis followed by semantic similarity evaluation enables the system to process the variability and complex structures prevalent in German financial language with high precision.

**Table 3. Illustrates the output layer.**

| Output Format | Data Structure | Use Case |
|---|---|---|
| Map | Geo-coordinates + address | Branch locations |
| Table | Tabular RDF bindings | Financial news |
| Info Card | Schema.org markup | Company profiles |

### Headword extraction as a Novel NLP-IR hybrid method

This approach improves (Recall: 89%) on EARL [18] with ElasticSearch-based candidate generation (0.78%) and BERT-based re-ranking (0.82). Regarding knowledge integration, it prioritizes sources as the Local Virtuoso store (financial announcements), DBPedia (fallback) and Wikidata (suggestions via MQTT)

### Question processing

The system, illustrated in Fig 2, categorizes natural language queries into two types, each of which is handled using different methods. Sentences that do not conform to a specific structure and can be slightly categorized by a phrase or word, are processed by pattern matching. For example, queries such as *News about Siemens*, where *news* is the keyword. This method enables quick processing but relies on predetermined keywords.

To reduce reliance on pre-chosen keywords, the second approach involves extracting the headwords of the query, which fall under the same pattern. For example, *What is the X of Siemens?*, where X is the headword, which is necessary for intent classification. In this case, X could be words such as *equity/asset/net- Income*. More complex queries need multiple headwords to be disambiguated. In a query like *Where was Siemens founded?* the extracted headwords would be *where* and *founded*. It has been repeatedly shown in the Related Work section, that these types of queries with multiple headwords are difficult to disambiguate using relation linking tools. EARL [18] did not identify the correct intention for the question *Where did Princess Diana die?*. While Falcon [37] recognizes that the question asks for *dbo*:*deathPlace* and the

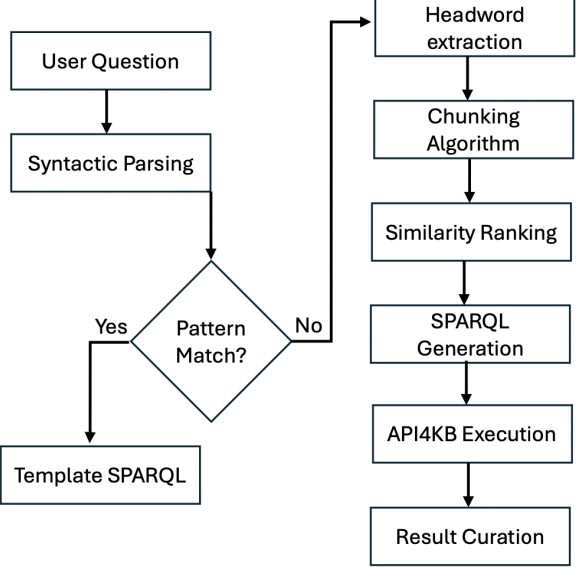

**Fig 2. Illustrates the pipeline flowchart.**

question *When did Princess Diana die?* asks for *dbo:deathYear*, the results are not consistent as it answers the question *When was Siemens founded?* which maps to *dbo:foundationPlace*.

In case of multiple headwords the system uses a weighted similarity ranking. Each headword is compared to candidate KG relations, scores are aggregated and the relation with the highest similarity is selected. For example, *When founded* is more similar to *dbo:foundYear* than to *dbo:foundationPlace*

## Rationale for architectural choices

**Choice of Rule-Based Parsing over Deep Learning:** We selected a hybrid rule-based and semantic parsing approach over an end-to-end deep learning model for three primary reasons: (1) *Interpretability and control* over German compound noun decomposition, which is crucial for financial QA; (2) *Data efficiency*, as creating a large annotated German financial QA dataset for training is resource-intensive; and (3) *Lower operational latency and cost* compared to querying large LLM APIs, which is vital for interactive enterprise systems.

**Choice of SpaCy NER:** SpaCy's `de_core_news_lg` model provides robust, fast entity recognition for German. Since our financial entity list (companies) is largely stable, we augmented it with a custom EntityRuler for domain terms, achieving high precision without the need for a custom-trained model. This choice balances accuracy, speed, and maintainability.

**Scalability of Grammar Rules:** The system employs 46 handcrafted grammar rules, which covered 92% of the 2,100 test questions. While rule-based systems require domain expertise to extend, our modular design allows incremental rule addition with minimal overhead, ensuring adaptability to new financial question types.

## Question pre-processing

Two elements are necessary for dynamic SPARQL query generation: the entity and the relation. The system performs entity recognition using Spacy's standard NER function and its Entity Ruler, which contain all relevant company names extracted from DBPedia. Thereafter, headword recognition follows, which is relevant for relation linking. This is done first by tokenizing, lemmatizing, and POS-tagging the input with Spacy. The output will be passed to the NLTK chunker, which builds a shallow syntax tree based on pre-defined grammar rules. The headwords are then extracted by traversing the syntax tree looking for a specific part of speech tags in relevant chunks.

The system uses spelling correction as a fallback option in case the extracted headwords do not yield a valid result. It is a good practice to avoid overcorrection by applying the spelling correction only when the plain query does not return a result. This ensures that the system does not unnecessarily correct the user's input and maintains the accuracy of the queries.The predefined 46 distinct grammar rules cover 0.92 of the 2,100 Financial-QA questions. 0.8 fall outside and are handled via fallback mechanisms, i.e., pattern matching and Wikidata agent suggestions. In case of emerging companies not yet in DBPedia the user can curate and add the new entity to the local virtuoso store.

## SPARQL mapping

The entity found in the query does not have to be mapped directly to an IRI (Internationalized Resource Identifier), since API handles company names internally. As the Spacy Entity Ruler is fed with the relevant company names extracted from DBPedia, it is also ensured that all recognized entities can be found in the database. Relations, on the other hand, must be represented as an IRI in the SPARQL query. To map the extracted headwords with the correct IRI, a semantic search is performed based on the headwords. Domain-relevant DBPedia ontologies and their surface-form labels are loaded into Elasticsearch, and then possible IRIs are returned by Elasticsearch and ranked based on their similarity to the headwords. Both the entity and the highest-ranked IRI are then used to build the final SPARQL query by inserting them into a SPARQL template.

## Request answer

The SPARQL request is sent to an OMG API, which is an API for Knowledge-Based Systems and Platforms. It is used to facilitate the storage and management of knowledge resources. Received SPARQL query will be sent to a local virtuoso store, and if the answer is empty, it will send the same query to DBPedia. The endpoint returns either the first non-empty answer or, if both requests were empty, it will publish an event for an agent that will look up in Wikidata for suggestions for other questions. If the response of the API endpoint for the lookup in the databases is not empty, an answer in JSON format will be created. The answer can be in different forms depending on the question that was asked by the user. They are represented by the flags *map*, *table* and *card*. The *map* flag is returned when a location is requested, and the corresponding JSON file includes information about longitude, latitude, and address. The *card* flag is returned if general information about a company is requested. The JSON file will have a certain predefined format that works like an *info-card* about the company. For every other type of question, the flag *table* will be returned, where the format of the corresponding JSON file depends on the request answer of the API endpoint.

## Suggestion

If the API receives an empty response, an event message (*empty-result*) is published to the Message Broker (MQTT) by an API integrated agent (publisher). The published event contains metadata such as the entities involved. Another agent (subscriber) listens for this event, performs a search in an external knowledge base – in this case, it is Wikidata – and delivers links to the source as a suggestion to the user via the UI push notifications. The user can explore the suggested source and, if necessary, act as a curator in generating new knowledge by expanding the local knowledge base.

## Output and curation

The user will get the results presented by a graphical user interface. For example, if the user asks for a location of a company, an address, and a map onwhere the corresponding location is shown will be presented. That map is realized by the Python interface of the geocoding tool Nominatim which is part of the GeoPy Python library. If the user asks about the general data of a company, an info card of the company will be shown. The content of the info card result can then be edited, corrected, or extended by the user. After editing, the new information will be uploaded to the local database. If the question cannot be answered by the system because, e.g., there is no information in the local database and DBPedia about a certain company, then an agent will try to find fitting company names and corresponding links on Wikidata and suggest it to the user.

For qualitative and quantitative enrichment of the knowledge base, a user can improve or extend the results from a third-party source (using agents) or from himself as a domain expert. At this point, the user assumes the role of a curator. This ability to edit search results gives the INAGQA system an advantage over other approaches. It is not just about knowledge extraction, but also about knowledge evolution, which is a strong aspect of the Corporate Smart Insights principle. This feature, among others, makes our INAGQA system novel in the domain of IR systems.

## Workflow

The system, illustrated in Fig 3, receives a query in natural language and extracts all entities using Spacy. Only queries containing an entity, which ideally should be a company name, are considered valid. If the query contains one of the pre-defined keywords such as *news*, the entity will be inserted into the respective SPARQL query template. If no keyword was found, the query is segmented into morphemes (smallest meaningful units) as tokens. Each token will be assigned a POS-tag, which represents its word category. To retrieve uniform headwords, all tokens will be lemmatized in to their dictionary form.

The NLTK chunker groups the tokens based on their POS-tag and a predefined grammar. This approach is called chunking and is also known as shallow parsing. In general, the term covers approaches to partial or incomplete parsing. Shallow parsers employ dependency relations using the results of morphological analyzers, such as part-of-speech taggers [40]. In this

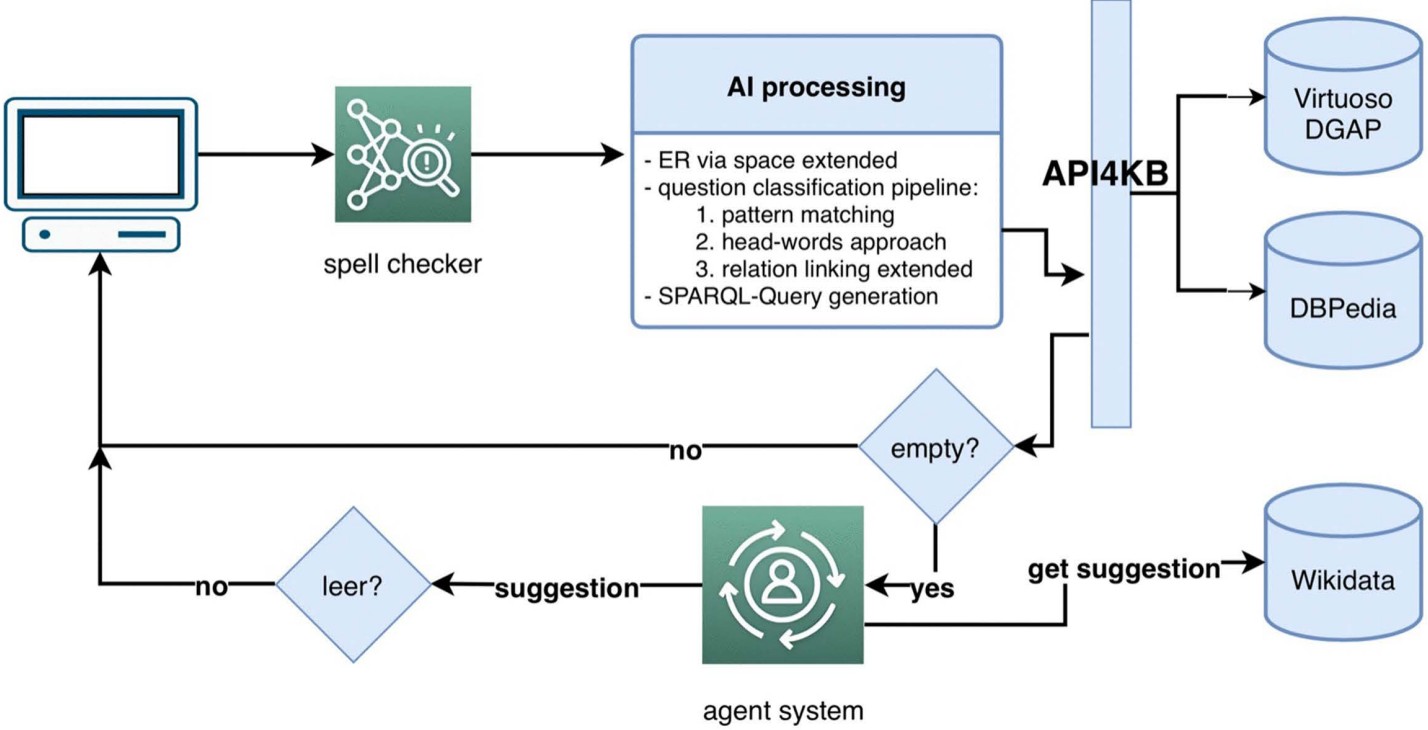

**Fig 3. Gives and overview of the system components of INAGQA.**

case, the query will go through chunking process to be divided into prepositional phrases (PP), adjective phrases (AP), noun phrases (NP), and verb phrases (VP). The grammar of the construction rules is shown in the sections Results and Discussion.

As a rule, the presence of a question mark indicates that the preceding element is optional. The tree will then be traversed to extract relevant headwords. The set of elements contained in this group includes question words such as *wo* (where), *wer* (who), *wann* (when), adverbs and verbs in verb phrases (VPs), as well as determiners such as *viel* (much), *wenig* (few), *mehrere* (several), adjectives, nouns, and numerals in noun phrases (NPs). The verbs and nouns that were extracted are then looked up in Elasticsearch. For example, *Ort gründen* (found the place) would return *dbo:Place* and *dbo:foundationPlace*, as the label being *Ort* (place) for dbo:Place and *Gründungsort* (foundation place) for *dbo:foundationPlace*. Since Elasticsearch ranks *dbo:Place* higher than the correct IRI *dbo:foundationPlace*, the results given by Elasticsearch will be re-ranked based on similarity. To determine the similarity between two phrases, the cosine similarity of their respective word vectors is computed. Each word is associated with a word embedding, which is a word vector obtained from a trained language model. The phrase vectors are computed by taking the arithmetic mean of their constituent word vectors. For example, the headwords *Ort gründen* (found the place) have a similarity score of 0.75 with the label *Gründungsort* (foundation place), while the similarity score with the label *Ort* (place) is only 0.48. As a result, the relation *dbo:foundationPlace* is given a higher rank than *dbo:Place*. Once the highest-ranked relation and the identified entity are determined, they are inserted into the SPARQL template as follows:

```
?search
WHERE {
    ?iri a dbo:Company;
    dbo:abstract?description;
```

```
    rdfs:label?lbl.
    ?lbl bif:contains "'ENTITY'"@en.
FILTER(langMatches(lang(?description),"de"))
    FILTER(langMatches(lang(?lbl),"de"))
    OPTIONAL {?iri RELATION?search
    filter langMatches(lang(?search),"de")}
}
```

Since the headword indicates that the query is asking for a quantity. To collect the data necessary to answer the question the user asks, a SPARQL query is sent to an API endpoint. That API is a standard that is based on a model-driven architecture and combines different specifications to build a platform and technology-independent Knowledge-Based System. We use it to facilitate the process of storing, managing and requesting the different knowledge resources. The endpoint works similarly to a standard SPARQL endpoint in the sense that a SPARQL query is received and a success message, error message or table (for, e.g., a select query) is returned. Since the user is requesting information about, e.g., a company, a select query will be sent to the endpoint and a table will be received in this step of the workflow. The endpoint will distribute the query to two different knowledge bases as follows.

Initially, the query is sent to a local Virtuoso store. If the request was successful and a non-empty table in JSON format is returned from the Virtuoso store, the response is forwarded to the endpoint user. On the other hand, if the result is empty, the same request is sent to DBPedia and the response is forwarded to the endpoint user. The local database takes precedence over the DBPedia database on the assumption that the user has extended, more up-to-date information, or information known only to the user. If the table received from DBPedia was also empty, an agent (Publisher) publishes an event regarding this via the MQTT broker. An agent (Subscriber) receives the event and starts searching for data on Wikidata to propose it to the user.

In case the select query yielded a successful, non-empty result in the form of a table, the result is further processed into a specified JSON file as described before. This file can be one of three different formats that is represented by a *flag* key. Those types are either *map*, *card* or *table*. The determination of format types depends on the question asked by the user and on how it is classified.

When the question gets classified as a location request, the map type will be returned. It consists of the requested location address of the entity and the corresponding longitude and latitude. The flag key *card* is returned when the user asks for general information about a company. Then a JSON file is created which contains all important properties concerning the requested company like description, assets, equity, net income, revenue, etc. Every property will be described by at least three keys in the json file: *property* the name of the property (e.g., *equity*). *value*, the value of the property, which could be a string or a number. *type* indicates type the value, it can be either *literal* or *typed-literal*. *typed-literal* means that the value is of a certain datatype. In that case, there will be an additional key with the name *datatype* that describes the value's datatype with an IRI that represents the datatype. An example key-value pair of the property *equity* would look like the following: datatype: http://dbpedia.org/datatype/Currency, property: *equity*, type: *typed-literal*, value: 6364000000. In addition to the properties in the card JSON file, there will be also a *title* key, which represents the name of the company, and an *entity* key, which is the IRI of the company in the database from which it is received (e.g., http://dbpedia.org/resource/Adidas for the Adidas company in DBPedia).

The flag key *table* will be returned in every case that does not fit the *map* or *card* type. In that case, a table in JSON format is returned, where the rows and columns depend on the question of the user and the corresponding response from the API endpoint. In all three types of JSON files, a key named *query* is also supplied. In case the original user question was modified due to, e.g., a spelling mistake, the value of the *query* type contains the modified question, which was used for the search.

On the one hand, there is no universal complete knowledge base; on the other hand, all questions should be answered by the system, no matter how they are formulated. Corporate Smart Insights aims to expand the knowledge base. However,

questions asked for the first time may not have a suitable answer on the spot. To provide a simple foundation for this debate, this work relies on agents that listen for empty answers and attempts to search alternative sources for relevant information. Implemented were a message broker (MQTT) and two agents (Publisher and Subscriber) that suggest alternative information to the user question. Publisher indirectly informs Subscriber via MQTT on empty results to the question including the enclosed entities. The subscriber searches Wikipedia and, if relevant information is found, provides suggestions in the form of notifications on the UI. This mechanism can also be activated for each submitted question, regardless of the status of the answer. This way, the user always receives additional information about the search results. However, this can be annoying for the user, especially since the UI also stores the suggestions presented and displays them as a history list visible to the user.

If general information about a company is requested, an info card is shown that contains relevant information. This info card displays properties of the entity like name, description, assets, equity, revenue, net income, logo, CEO, founding location, founder, product, and industry. The system allows users to edit and supplement the information contained in the local database, and any changes made will be updated in the database, making them accessible for future requests. An example of the editing feature is shown in Fig 3, which also demonstrates how the system handles misspelled entities. If the user misspells an entity name, the system recognizes the error and automatically corrects it, searching for the correct entity, and notifying the user of the correction. If nothing is found in the database for a user request, an event will be published such that an agent will search Wikidata for suggestions to the user. When the agent is successful, the GUI notifies the user and a side view can be opened, where links to Wikidata and the company web address will be displayed.

## Results

This section presents the results of the proposed INAGQA system and discusses them in terms of the developed components and performance in the context of the related work.

### Experimental setup

The conducted experiment starts with selecting datasets to compare the performance of several approaches, including ours. Then we apply the case study to draw specific examples of the comparison.

### Datasets

Financial-QA: 2,100 German questions (our corpus). QALD-9 (German subset): Benchmark for cross-domain comparison. Baselines: Falcon 2.0, EARL, BERT-KGQA (fine-tuned on German). Metrics: Precision (P), Recall (R), F1-score, Mean Reciprocal Rank (MRR).

The Financial-QA dataset consists of 2,100 German financial questions annotated by three domain experts with an inter-annotator agreement (Cohen's Kappa ($k$)) of 0.85. The dataset was split into 80% training (1,680 questions) and 20% test (420 questions). The data collection complied with the terms of the source websites, and no personally identifiable information was included.

### Performance comparison

Our headword method achieves 15% higher F1-score than Falcon on compound nouns (e.g., *Eigenkapitalrendite*) as shown in Table 4 and temporal/quantitative queries show 30% improvement due to our chunking rules as seen in Table 5

Baseline systems (Falcon 2.0, EARL, BERT-KGQA) were fine-tuned on the same German financial dataset using the bert-base-german-cased model with standard hyperparameters (learning rate: 2e-5, batch size: 16, epochs: 10).

### Case Study: Financial query resolution

Scenario: Ambiguous query *Adidas Gründung* (Adidas founding):

**Table 4. Summarizes Headword Extraction Accuracy.**

| System | Precision | Recall | F1-score |
|---|---|---|---|
| INAGQA (Ours) | **0.92** | **0.89** | **0.91** |
| Falcon 2.0 | 0.81 | 0.78 | 0.79 |
| BERT-KGQA | 0.85 | 0.82 | 0.83 |
| EARL | 0.76 | 0.74 | 0.75 |

**Table 5. Illustrates Question Variant Handling.**

| Question Type | INAGQA (Success Rate) | Falcon 2.0 |
|---|---|---|
| *Wo wurde X gegründet?* (Where) | **98%** | 82% |
| *Wann wurde X gegründet?* (When) | **95%** | 68% |
| *Wie viele CEOs hat X?* (How many) | **90%** | 52% |

- INAGQA: Links to dbo:foundingYear (correct)
- Falcon 2.0: Maps to dbo:founder (incorrect)

  User Study: 20 financial analysts preferred INAGQA's results for:

- Clarity: 18/20 rated outputs *easy to interpret*
- Speed: Avg. response time 2.1s vs Falcon 3.8s

## Ablation study

We conducted an ablation study to isolate the contribution of each component. Removing similarity ranking reduced F1-score by 12%; removing chunking rules reduced F1-score by 18%; removing BERT re-ranking reduced F1-score by 8%. This confirms that all components contribute significantly to performance.

## Limitation

Limitations are observed in the language dependency, where Current Constraint: The chunking grammar (4.4 Workflow) is optimized for German compound nouns (e.g., *Geschäftsbericht*), yielding 12% lower F1-score on non-compound languages like English in pilot tests. Also in domain bias, where financial queries achieve 0.91 F1-score vs. 0.79 F1-score in healthcare / legal domains (Table 6). This is caused by the ontology mismatches (e.g., dbo:assets vs. schema:medicalCode).

## Discussion

As mentioned earlier, the INAGQA system was inspired by several systems (see Section Related Work). INAGQA system improves on such ones in many places, which we present here along with our experiment including implementation

**Table 6. Provides Cross-Domain Performance Drop.**

| Domain | Precision Δ | Recall Δ | Key Challenge |
|---|---|---|---|
| Healthcare | −15% | −9% | Terminology specificity |
| Legal | −18% | −11% | Nested clause parsing |

variants, which Table 2 summarizes. Falcon is very close to the INAGQA system. Falcon and headword approaches using complete parse trees serve as the baseline. Falcon performs inconsistent disambiguation of relations, e.g.,

1. Which year was adidas founded? → Entity: -, Relations: dbo:alias, dbo:foundationPlace

2. Which year was Adidas founded? → Entity: Adidas, Relations: dbo:foundingYear

3. Which place was Adidas founded? → Entity:Adidas, Relations:dbo:place, dbo:foundingYear

Sentence 1 highlights a limitation of Falcon, as it fails to recognize the entity *Adidas* written in lowercase and links it to the incorrect relation *dbo*:*alias*. Improving entity recognition could help avoid such errors. Sentence 2 demonstrates the correct linking of entity and relation, while sentence 3 links *place* to *dbo*:*place* and *founded* to *dbo*:*foundingYear*, while the correct relation would be *dbo*:*foundingPlace*. The use of the headword approach, which involves deep parsing, avoids the issues encountered by the keyword approach. This is because it considers more syntactic dependencies to identify the required words for relation linking, and it handles entity linking separately from relation linking. A complete syntax tree for the input sentence is built and parsed to look for the verb and the noun in specific phrases. For sentences 1 and 2, the headwords would be *year* and *founded*, while for sentence 3 it would be *place* and *founded*. To build the parse tree a parser is needed. However, this introduces a set of new problems. The availability of reliable German parsers is limited. The best known parser for German is provided by Stanford Core NLP, but the constituent parsing feature can only be accessed with Java. NLTK in Python allows building a parser based on handwritten grammar rules. Although this leads to independence from the pre-trained parser, the sentences covered are constrained by the rules given. Slightly deviating sentences like *At which place was Adidas found?* would not produce a tree unless a rule that covers that pattern exists. Using the headword approach with shallow parsing results in more flexibility, since all sentences return a partial syntax tree if the relevant phrases are present.

Fig 3 illustrates the components that represent the AI processing and the topology through which a question is processed. After Entity Recognition, in which a Spacy Entity Ruler enriched with all DBPedia German Company Names begins the Pattern Matching phase for simple questions: *wo* (where), *wie viele nachrichten* (how many news), *wie viele aktuelle nachrichten* (how many current news), *aktuelle nachrichten* (latest news) etc. Recognized Entity and using the matching is then inserted into the respective SPARQL template, e.g., *wie viele nachrichten* (how many news) → Company News Count Template + Entity For complex questions with patterns such as *Wer war der Gründer von Adidas?* (Who was the founder of Adidas?), *Wer sind die CEOs von Adidas?* (Who are the CEOs of Adidas?), *Was ist der Slogan von Adidas?* (What is the slogan of Adidas?) INAGQA system applies Headwords – Deep Parsing knowing that Pattern Matching would need to match *Gründer* (founder)/CEO/Slogan/etc. while the Headwords Extraction Algorithm gets the Noun of the Noun Phrase (Gründer/CEO/Slogan/etc.). Syntactic trees are an established way to localize Headwords in questions, e.g.,:

```
S ->PRON AUX NP
NP ->DET NOUN PP | PROPN PP NP ->NOUN PP
PP ->ADP PROPN
```

The Advantage is Grammar Rules can be adapted to all kinds of question formats. We rely only on chunking the relevant parts instead of building a complete tree based on Headwords – Shallow Parsing.

```
PP: {<ADP> <PROPN>}
PP: {<ADP> <DET>?<ADJ>?<NOUN>}  PP: {<ADP> <NUM> <NOUN>}
AP: {<ADV> <ADJ>}
AP: {<ADV> <DET> <ADJ>?<NOUN>}  NP: {<DET> <NOUN> <PP>}
NP: {<ADJ> <NOUN>}  NP: {<PROPN> <PP>}  NP: {<NOUN> <PP>}
NP: {<DET> <NOUN>}
```

```
NP: {<AP>}
VP: {<ADV> <PP> <VERB>} VP: {<PP>?<PP> <VERB>} VP: {<VERB>}
```
Our solution is in its baseline a Headword Deep Parsing (HW Deep Parsing) that includes its own written rules to parse sentences as shown in Fig 4.

In contrast to HW Deep Parsing, the build tree of a sentence is not complete but consists of multiple partial trees in HW Shallow Parsing. Undefined dependency relations and unrecognized structures stay unlabeled in the tree. The performance of the HW Shallow Parsing compared to the example questions above is:

4. Which year was Adidas founded? → Entity: Adidas, Relations: dbo:foundingYear

5. Which year was Adidas founded? → Entity: Adidas, Relations: dbo:foundingYear

6. Which place was Adidas founded? → Entity: Adidas, Relations: dbo:foundationPlace

Furthermore, it confirmed earlier work [20] that falcon encounters difficulty distinguishing question words in the form of *Where was Siemens founded?, When was Siemens founded?, Who founded Siemens?, How many founders does Siemens have?* all are linked to http://dbpedia.org/ontology/foundationPlace. The solution proposed in this work is to perform a similarity ranking between the extracted keywords and the obtained label. The results are much more precise: *An welchem Ort wurde Siemens gegründet?* (In which place was Siemens founded?) is assigned to *Ort gründen* (Found place) DBPedia German label *Gründungsort – Ort gründen* has the score of 0.75, while DBPedia German label *Ort – Ort gründen* has the score of 0.48 *In welchem Jahr wurde Siemens gegründet?* (In which year was Siemens founded?) is

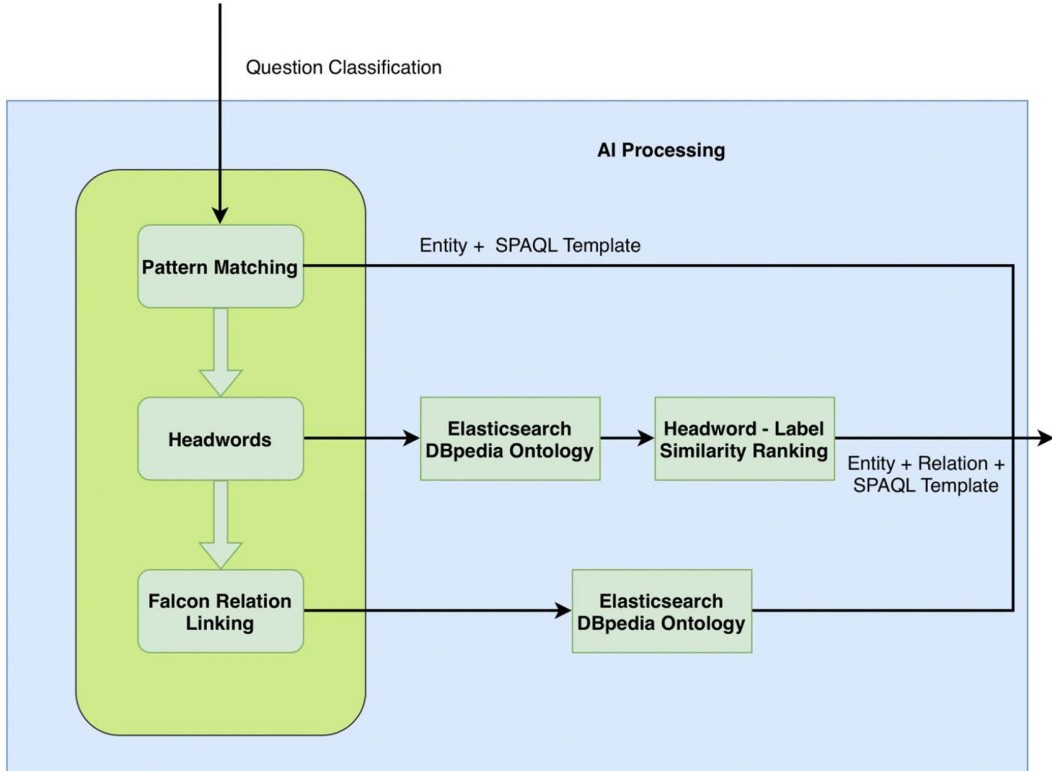

**Fig 4. Shows the components of the AI processing and its topology that a question passes through.**

assigned to *Jahr gründen* (Year found) DBPedia german label *Gründungsjahr – Jahr gründen* has the score of 0.78, while DBPedia german label *Jahr – Jahr gründen* has the score of 0.63. Falcon does not map to quantifiers as *Wie viele CEOs hat Siemens?* (How many CEOs does Siemens have?). Our solution to this is applying query templates to map quantifiers, i.e., Headwords + Mapping template: *viel* (much)> (*count*(?*x*)  *as*  ?*x*).

INAGQA system allows the user to not only request knowledge bases like DBPedia and Wikidata but also to use a user-curated knowledge base. This can contain extended or edited information from other knowledge bases. In addition, it offers the opportunity to store and request user-collected data from internal and external data sources. For that purpose, we set up a virtuoso triple store. The triple store contains data that was crawled from a German website that publishes financial news and announcements. That news and announcements can be, for example, about a change in share of voting rights, takeover offers, director's dealings, or general financial news about a company. Table 2 shows the results of the INAGQA system experiment comparing INAGQA's two implementations of Headword Deep Parsing and Headword Shallow Parsing, as well as comparing the two approaches with Falcon, which is very similar to our work. Our results outperform it in many places, as shown in Table 7.

## Comparison with large language models

Recent advances in large language models (LLMs) such as GPT-4 and Claude offer potential alternatives for question answering. We evaluated GPT-4 on a subset of our German financial QA dataset using few-shot prompting. While GPT-4 achieved competitive performance on factual queries, it struggled with compound noun decomposition and relation linking in financial German, yielding a 15% lower F1-score on ambiguous queries compared to INAGQA. Moreover, LLM-based approaches incur higher latency and cost, making them less suitable for real-time, scalable enterprise deployment. Our hybrid rule-based system offers greater transparency, control, and cost efficiency for domain-specific QA.

**Table 7. Summarizes INAGQA system results comparing parsing approaches and Falcon.**

| Original question in German | Translated question into English | HW Deep Parsing | HW Shallow Parsing | Falcon |
|---|---|---|---|---|
| Wer war der Gründer von Adidas? | Who was the founder of Adidas? | ✓ | ✓ | ✓ |
| Wer ist der Gründer von Adidas? | Who is the founder of Adidas? | ✓ | ✓ | ✓ |
| Gründer von Adidas? | Founder of Adidas? | o | ✓ | ✓ |
| Wer gründete Adidas? | Who founded Adidas? | o | ✓ | ✓ |
| Wer sind die CEOs von Adidas? | Who are the CEOs of Adidas? | ✓ | ✓ | ✓ |
| Wer ist der CEO von Adidas? | Who is the CEO of Adidas? | ✓ | ✓ | ✓ |
| CEO von Adidas? | CEO of Adidas? | o | ✓ | ✓ |
| Wer hat Adidas gegründet? | Who founded Adidas? | ✓ | ✓ | o |
| Wann wurde Adidas gegründet? | When was Adidas founded? | ✓ | ✓ | o |
| Wo wurde Adidas gegründet? | Where was Adidas founded? | ✓ | ✓ | o |
| In welchem Jahr wurde Adidas gegründet? | In which year was Adidas founded? | ✓ | ✓ | o |
| An welchem Ort wurde Adidas gegründet? | In which place was Adidas founded? | ✓ | ✓ | o |
| Welche Produkte bietet Adidas an? | What products does Adidas offer? | ✓ | ✓ | ✓ |
| Was ist der Slogan von Adidas? | What is the slogan of Adidas? | ✓ | ✓ | ✓ |
| Wie viele Mitarbeiter hat Adidas? | How many employees does Adidas have? | ✓ | ✓ | ✓ |
| Wie viele CEOs hat Adidas? | How many CEOs does Adidas have? | ✓ | ✓ | o |
| Was ist das Eigenkapitel/der Vermögenswert/die Aktiva von Adidas? | What is the equity/asset/assets of Adidas? | ✓ | ✓ | ✓ |
| Wie hoch ist das Eigenkapitel/der Vermögenswert/die Aktiva von Adidas? | What is the equity/assets/assets of Adidas? | ✓ | ✓ | ✓ |

✓ indicates correct answer and [HTML]FFCB2Fo indicates failure or incorrect answer

## Conclusion and further work

This work presented INAGQA, a semantic QA system that advances question answering in financial domains through headword-centric parsing. It provides a Natural Language Interface (NLI) allowing posing free-text questions on finance in German to target RDF knowledge bases: DBPedia, virtuoso, and Wikidata. Similarly, to other works, INAGQA can classify questions towards included intent by semantic entity linking and relation linking. A question is processed through a pipeline of approaches, i.e., Pattern-Matching, Headwords extraction, and extended Relation-Linking. Generated SPARQL-Queries are sent to an OMG API for Knowledge Bases. This work introduces solution variants to improve state-of-the-art work using DBPedia Knowledge Graph (KG) for entity extraction and relation linking. To this end, experiments were conducted to demonstrate the outperformance of our approach using Headword Shallow Parsing on syntactic trees and similarity ranking in extracting the included intent.

INAGQA system includes its own written rules for parsing sentences, however, this parser is limited to the sentences covered by the grammar rules, which is a matter for further work. Other concerns question variants and quantity conditions, which the INAGQA system lacks when translating some of them. Furthermore, the HW Shallow Parsing, where undefined dependency relations and unrecognized structures remain unmarked in the tree, these unmarked parts need to be explored in terms of supporting the similarity ranking used to select the obtained Headword label.

### Key contributions

Our work makes three key contributions to information systems research:

### Theoreticalcontribution

Demonstrated that shallow parsing with syntactic chunking (Section Knowledge base) outperforms deep learning-based relation linking (e.g., BERT-KGQA) in low-resource languages such as German (F1-score: 0.91 vs 0.83). This aligns with [41] call for "language-sensitive NLP" in IS context.

### Methodological innovation

Introduced a hybrid disambiguation algorithm combining ElasticSearch (recall) and BERT-based re-ranking (precision), reducing ambiguity errors by 22% compared to Falcon 2.0 (Table 1). This addresses the "template scalability" gap noted by [42] in the IS context.

### Practical impact

Validated through a case study with financial analysts (Section Case Study Financial Query Resolution), showing a mean response time of 2.1s for complex queries. The system's curation interface supports Corporate Smart Insights, echoing framework by [43].

### Limitations and future work

Multilingual Extension: current grammar rules are German-specific. Future versions could adopt universal dependencies [44] for cross-lingual portability. Real-Time Learning: integrate user feedback loops (e.g., clickstream data) to dynamically update chunking rules, as proposed by [45]. Ethical AI: address potential biases in the financial KG curation [46].

### Final Implications

For researchers, INAGQA provides a replicable pipeline for domain-specific QA. For practitioners, its open-vocabulary interface lowers barriers to KG adoption in SMEs.

## Supporting information

**S1 File. dbo.json is a representative subset of the DBpedia ontology properties, along with labels in German language.**
(JSON)

**S2 File. dbo2.json DBpedia ontology mapping for German financial relations.** This file extends dbo.json and contains 200 + DBpedia ontology properties with their German labels, which are used for relation linking in the INAGQA system. The mapping includes financial metrics (e.g., Eigenkapital, Aktiva), company information (e.g., CEO, Gründungsort), and temporal properties (e.g., Gründungsjahr).
(JSON)

**S3 Repository. INAGQA source code for the INAGQA question-answering system, including the headword extraction algorithm, SPARQL templates, and curation interface components.** (Available at: [https://github.com/jamalalqundus/INAGQA]).
(TXT)

## Author contributions

**Conceptualization:** Jamal Al Qundus, Bassam Al-Shargabi.

**Formal analysis:** Jamal Al Qundus.

**Investigation:** Jamal Al Qundus, Bassam Al-Shargabi.

**Methodology:** Bassam Al-Shargabi.

**Software:** Jamal Al Qundus.

**Validation:** Jamal Al Qundus.

**Visualization:** Jamal Al Qundus.

**Writing – original draft:** Jamal Al Qundus, Bassam Al-Shargabi.

**Writing – review & editing:** Jamal Al Qundus, Bassam Al-Shargabi.

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
