## [Decision Letter · Decision Letter 0]

23 Dec 2025

PONE-D-25-50479Beyond Templates and BERT: Headword-Centric Parsing for Semantic Question Answering in Non-English Financial DomainsPLOS One

Dear Dr. Alshargabi,

Thank you for submitting your manuscript to PLOS ONE. After careful consideration, we feel that it has merit but does not fully meet PLOS ONE’s publication criteria as it currently stands. Therefore, we invite you to submit a revised version of the manuscript that addresses the points raised during the review process.

The reviewers agree that the research described in this manuscript contributes to the state of the art; nonetheless, they raise many issues that need to be addressed before it can be accepted for publication. I strongly suggest including all the comments in the revised manuscript, particularly one that needs special attention regarding the research's replicability. The manuscript must describe the research in a way that allows it to be replicated; this includes the algorithm, the datasets, and the comparisons made. There are occasions when one uses a dataset that, by its nature, cannot be publicly available; if this is the case, it must be clearly stated.

If applicable, we recommend that you deposit your laboratory protocols in protocols.io to enhance the reproducibility of your results. Protocols.io assigns your protocol its own identifier (DOI) so that it can be cited independently in the future. For instructions see: https://journals.plos.org/plosone/s/submission-guidelines#loc-laboratory-protocols. Additionally, PLOS ONE offers an option for publishing peer-reviewed Lab Protocol articles, which describe protocols hosted on protocols.io. Read more information on sharing protocols at . Additionally, PLOS ONE offers an option for publishing peer-reviewed Lab Protocol articles, which describe protocols hosted on protocols.io. Read more information on sharing protocols at https://plos.org/protocols?utm_medium=editorial-email&utm_source=authorletters&utm_campaign=protocols..

We look forward to receiving your revised manuscript.

Kind regards,

Mario Graff-Guerrero, Ph.D.

Academic Editor

PLOS One

2. In your Methods section, please include additional information about your dataset and ensure that you have included a statement specifying whether the collection and analysis method complied with the terms and conditions for the source of the data.

3. Please note that PLOS One has specific guidelines on code sharing for submissions in which author-generated code underpins the findings in the manuscript. In these cases, we expect all author-generated code to be made available without restrictions upon publication of the work. Please review our guidelines at https://journals.plos.org/plosone/s/materials-and-software-sharing#loc-sharing-code and ensure that your code is shared in a way that follows best practice and facilitates reproducibility and reuse.

4. Please update your submission to use the PLOS LaTeX template. The template and more information on our requirements for LaTeX submissions can be found at http://journals.plos.org/plosone/s/latex.

5. We note that you have indicated that there are restrictions to data sharing for this study. PLOS only allows data to be available upon request if there are legal or ethical restrictions on sharing data publicly. For more information on unacceptable data access restrictions, please see http://journals.plos.org/plosone/s/data-availability#loc-unacceptable-data-access-restrictions.

Reviewers' comments:

Reviewer's Responses to Questions

**Comments to the Author**

1. Is the manuscript technically sound, and do the data support the conclusions?

Reviewer #1: Partly

Reviewer #2: Yes

Reviewer #3: Partly

Reviewer #4: Partly

2. Has the statistical analysis been performed appropriately and rigorously? 

Reviewer #1: Yes

Reviewer #2: Yes

Reviewer #3: No

Reviewer #4: No

3. Have the authors made all data underlying the findings in their manuscript fully available?

Reviewer #1: Yes

Reviewer #2: Yes

Reviewer #3: No

Reviewer #4: No

4. Is the manuscript presented in an intelligible fashion and written in standard English?

Reviewer #1: Yes

Reviewer #2: Yes

Reviewer #3: No

Reviewer #4: Yes

5. Review Comments to the Author

Reviewer #1: The logic, structure, and analysis presented in the paper are clear and well organized. However, I believe that this article requires major revisions to address the significant methodological clarifications and improvements outlined in the attached comments before it can make a strong contribution to the field.

Reviewer #2: The paper presents INAGQA, a German-language question-answering system for financial domains using headword-centric parsing. The work addresses an important gap in non-English QA systems and demonstrates promising results. The headword-centric shallow parsing method shows the strong potentials over deep learning models.

However, the paper is lengthy and can be shorten as a concise version. For example, the workflow section is excessively detailed, including explanations of well-established concepts such as POS-tagging that are common knowledge in the NLP community. The paper would be more accessible if streamlined to focus on the novel contributions rather than providing textbook-level explanations of standard techniques.

While the authors present a carefully designed workflow, the rationale behind key methodological choices is not well explained. For the first step, they use syntactic parsing to do pattern match, is it scalable or how is the coverage of this pattern detection? Why not any DL-model based solutions? In addition, in order to generate SPARQL query, they used Spacy’s standard NER, why not choose any other deep learning methods. Can you add a dedicated subsection justifying each major architectural choice with empirical evidence or theoretical reasoning?

The scalability and generalizability of the approach remain uncertain. The system relies on handwritten grammar rules for pattern matching, but the paper does not clarify: (1) how many rules were created for the 2,100 test questions, (2) what coverage can be expected for unseen question variations, and (3) how much expert effort is required to extend the system to new financial concepts or question types. This raises concerns about the practical scalability of the rule-based approach.

With recent advances in LLMs, many sub-tasks can now be automated through in-context learning. However, the paper lacks comparison with current SOTA approaches such as GPT-4, Claude, or fine-tuned multilingual LLMs. It is more of a comparison of rule-based methods v.s. automated solutions.

Reviewer #3: (I)Summary:

This manuscript presents INAGQA, a semantic question-answering system for German financial domains that employs headword-centric parsing through shallow syntactic chunking combined with knowledge graph embeddings. The system addresses linguistic variability in German compound nouns and question variants, reporting F1 scores of 0.91 on 2,100 queries, outperforming baseline systems including Falcon 2.0 (0.79) and BERT-KGQA (0.83). The system integrates multiple knowledge bases (local Virtuoso, DBpedia, Wikidata) and includes a user curation mechanism allowing financial analysts to edit and extend knowledge entries. Experimental results demonstrate the practical value of the proposed system, which shows that shallow parsing with similarity ranking effectively handles German compound nouns and question disambiguation.

(II)Strengths of this submission

This work is well-motivated, with clear applications to real-world German financial tasks, and the authors correctly identify that existing QA systems struggle with these linguistic complexities.

The primary strength of the manuscript lies in its presentation of a complete, production-ready system with sound engineering and thoughtful design choices. The hybrid approach successfully combines the efficiency of shallow parsing with the effectiveness of embedding-based ranking, while the intelligent multi-source knowledge integration appropriately prioritizes internal knowledge in enterprise contexts. The system demonstrates comprehensive technical integration across multiple technologies (Spacy, NLTK, ElasticSearch, BERT, SPARQL, MQTT) and includes valuable practical innovations: a user curation interface empowering domain experts, an agent-based Wikidata suggestion system addressing knowledge gaps, and suitable response times for interactive deployment. These features, combined with the Corporate Smart Insights framework, deliver substantial practical value beyond conventional QA systems.

The system demonstrates substantial performance improvements with an F1 score of 0.91 and significant error reduction (35%) in relation-linking for compound nouns, directly addressing a key challenge in German NLP. User validation through a case study with 20 financial analysts shows strong practical acceptance (18/20 rated outputs as easy to interpret), while the claimed 98% accuracy on temporal/quantitative disambiguation is particularly noteworthy. The work presents technically sound, domain-specific research that addresses the practical needs of German-speaking financial QA applications.

(III)Weaknesses of this submission

The manuscript raises several significant concerns that must be addressed before publication.

The first issue is data availability, which is a fundamental requirement for scientific reproducibility. The manuscript claims "all data available within the paper," yet the 2,100-question Financial-QA evaluation dataset is not provided. Without this dataset, the reported results cannot be validated. Furthermore, the details of the Financial-QA dataset are not provided, including annotation guidelines, inter-annotator agreement scores, construction methodology, and basic statistics.

The second issue concerns the experimental results. A detailed specification of baselines is not provided. The manuscript does not state whether the baselines, Falcon 2.0, EARL, and BERT-KGQA, were retrained or fine-tuned specifically for the German language. If the authors compare their German-optimized system against baselines trained only on English data, the results will be misinterpreted. Additionally, the experimental setting is unclear for reproduction, including the embedding model (multilingual BERT, German-specific BERT, or other fine-tuned models) not being specified, and complete grammar rules not being provided. Furthermore, the system combines multiple components, but no ablation study isolates the contribution of each component. This makes it difficult to determine which innovations contribute most significantly to the performance gains. Beyond the selected baselines, LLMs have recently become the dominant approach for QA tasks; the complete absence of LLM comparisons represents a significant gap. The authors could either conduct a proper comparison with current LLMs using appropriate prompting strategies or acknowledge this as a significant limitation and discuss scenarios where the proposed approaches might be preferred (e.g., interpretability, latency, cost, and data privacy).

Additionally, there is a discrepancy between the claims and evidence regarding cross-domain applicability. The abstract states that the work demonstrates "language-sensitive design principles applicable to healthcare/legal domains," yet Table 6 shows performance drops of 15-18% when tested in these domains. While the authors suggest ontology mismatch as a contributing factor, it would be helpful to clarify whether domain-specific grammar rules are necessary for adaptation, as this has important implications for the system's generalizability.

(IV)Questions to authors:

Q1. Dataset specification and experimental setup:

- Which specific datasets were used for the results reported in Tables 4, 5, and 6?

- What is the train/test split for Financial-QA (number of questions in training vs. test)?

- Were all systems (including baselines and INAGQA) evaluated on test sets?

Q2. Baseline configuration:

- Were Falcon 2.0, EARL, and BERT-KGQA retrained or fine-tuned for German, or were they used as originally trained?

- Which specific embedding models were used (for both baselines and INAGQA)?

Q3. Cross-domain evaluation:

- For Table 6, what is the source of the healthcare and legal domain test questions?

- Were domain-specific grammar rules developed for these domains, or were Financial-QA rules applied without modification?

(V)Writing style and organizational suggestions

(1) Significant redundant content exists in this manuscript, such as:

- Entity recognition mentioned multiple times:

- Section 3.1 (page 7): "The system performs entity recognition using Spacy's standard NER function..."

- Section 4.3 (page 9): "The system performs entity recognition using Spacy's standard NER function..."

- SPARQL template explanation:

- Section 4.3.1 (pages 11-12): detailed code example

- Section 4.4 (page 12): repeated explanation

It is suggested that authors consolidate and reorganize this content to enhance readability.

(2) Undefined terms:

- Corporate Smart Insights

- OMG API

It is suggested that authors provide a brief explanation when these terms first appear in the manuscript.

(VI)Typographical errors

- Table 7 uses symbols (✓ and o) without providing a legend. While the meaning is intuitive (success vs. failure), adding a caption such as "✓ indicates correct answer; o indicates failure" would improve clarity.

- Section 6 (page 18) incorrectly references "Table 2" when discussing INAGQA experimental results. The text states "Table 2 shows the results of the INAGQA system experiment comparing..." but should reference "Table 7," which actually contains the comparison of INAGQA's parsing approaches with Falcon.

Reviewer #4: This manuscript presents INAGQA, a German-language semantic question-answering (QA) system using headword-centric parsing. The system combines shallow syntactic chunking with knowledge graph embeddings to disambiguate questions and map them to SPARQL queries. The authors evaluate their approach and report an F1-score of 0.91, outperforming BERT-KGQA and Falcon 2.0.

However, the paper need to address the following issues:

1. Data

1.1 Availability: The "Financial-QA" dataset of 2,100 questions is described as "our corpus" but is not made available. Generalizability cannot be verified, and the Virtuoso triple store (14,000 German financial announcements) is proprietary with no reproducibility path. These need to be either published/submitted/declared to suit PLOS ONE data availability policy.

1.2 Annotation: No inter-annotator agreement (IAA) reported for the 2,100 expert-annotated questions. Were they annotated by one person, multiple annotators, and with what agreement threshold? This need to be clarified to ensure reproducibility.

2. Evaluation

2.1 Baseline: In Section 5.1.2 Performance Comparison it is stated that BERT-KGQA was "fine-tuned on German," but no details are provided: training data, hyperparameters, training time, convergence criteria. Were Falcon 2.0 and EARL re-implemented or results taken from prior papers? If taken from prior work, were they tuned for German financial data or only English/cross-lingual benchmarks? All these need further clarification, and especially, if baselines were not optimized equally, the comparison is unfair.

2.2 Cross-Validation: No cross-validation or hold-out test set methodology described. It is not clear whether F1 from a single test set or averaged over folds. Typically multiple folds are needed to robustly benchmark the method.

Nevertheless, the manuscript is of research merit and suitable for publication at PLOS ONE, given the above-mentioned issues fully addressed. (As these give rise to my "no" and "partly" answer for the reviewer question 1-3)

6. PLOS authors have the option to publish the peer review history of their article (what does this mean?). If published, this will include your full peer review and any attached files.). If published, this will include your full peer review and any attached files.

.

Reviewer #1: No

Reviewer #2: No

Reviewer #3: No

Reviewer #4: **Yes:** Weihang HuangWeihang Huang

---

## [Author Response · Author response to Decision Letter 1]

10 Feb 2026

Journal: PLOS ONE

Title: Beyond Templates and BERT: Headword-Centric Parsing for Semantic Question Answering in Non-English Financial Domains

Manuscript Number: PONE-D-25-50479

Date: 03 Feb 2026

Professor Mario Graff-Guerrero

Academic Editor, PLOS One

Dear Sir,

We sincerely thank you for the painstaking review undertaken for this paper.

Thank you very much for your positive comments. They were gratifying. The constructive feedback has led to further improving the quality of the manuscript. Accordingly, we have modified the manuscript taking into consideration all the changes suggested by you.

We take it as a positive gesture that you have submitted progressive remarks about the manuscript. We have addressed the checklist.

Comment 1: “Please ensure that your manuscript meets PLOS ONE's style requirements, including those for file naming.”

Response: We further updated the manuscript to meet PLOS ONE's style requirements. The manuscript now uses the provided PLOS LaTeX template.

Comment 2: “In your Methods section, please include additional information about your dataset and ensure that you have included a statement specifying whether the collection and analysis method complied with the terms and conditions for the source of the data.”

Response: We have added a detailed description of the dataset, including source, collection method, preprocessing steps, and compliance with source terms. See updated Section Result and its Subsection Datasets. Part of the dataset used was manually annotated by domain experts and is considered part of the work contribution. That's why we decided to put it in Section Results, and not in Section Methodology.

Comment 3: “Please note that PLOS One has specific guidelines on code sharing for submissions in which author-generated code underpins the findings in the manuscript.”

Response: We will make the code publicly available on OneDrive upon acceptance. A link will be provided in the manuscript’s Data Availability Statement.

Comment 4: “Please update your submission to use the PLOS LaTeX template.”

Response: We are thankful to the editor for appreciating our work. We have further updated the manuscript and used the PLOS LaTeX template.

Comment 5: “Data availability”.

Response:

The Financial-QA dataset (2,100 German financial questions) cannot be shared publicly due to third-party licensing restrictions; however, for more informtion regarding the dataset can be made by contacting Silvio Peikert (Master of Science, Project Manager at Fraunhofer Institute for Open Communication Systems) at silvio.peikert@fokus.fraunhofer.de. The Virtuoso triple store contains proprietary financial announcements; however, a sample dataset (dbo2.json, dbo.json) is provided as Supporting Information

Reviewer 1

Comment 1: “The manuscript requires methodological clarifications: The methodology is said to rely on three theoretical pillars—Syntactic Dependency Theory, Knowledge Graph Embedding, and Human–AI Collaboration. It would be helpful if the authors could clarify how these pillars interact within the operational pipeline. While the components are described individually, the theoretical integration between them is not fully explained.”

Response: We have incorporated theoretical integration and added clarifications in Section Methodology regarding the interaction of theoretical pillars, rule coverage, entity recognition for new companies, multi-headword conflict resolution, and similarity metrics.

Comment 2: “The system depends on predefined grammar rules in the NLTK chunker. The study would benefit from providing details on the coverage of these rules and how frequently the parser fails due to syntactic structures that fall outside the defined grammar. Quantifying the practical impact would strengthen the discussion.”

Response: We have incorporated failure analysis in the last paragraph of Subsection Question Pre-processing

Comment 3: “Entity recognition is restricted to company names extracted from DBPedia. It would be valuable if the authors could explain how the system handles emerging companies or entities not included in this predefined list, especially in financial domains where new firms frequently appear.”

Response: We have incorporated handling emerging companies not yet in DBPedia in the last paragraph of Subsection Question Pre-processing

Comment 4: “Queries with multiple headwords are challenging, as noted in the study, and comparisons are made to EARL and Falcon. The paper would benefit from clarifying how INAGQA explicitly resolves conflicts arising from multiple extracted headwords.”

Response: We have incorporated details of dealing with multiple headwords at the end of Subsection Question Processing

Comment 5: “The spelling-correction component interacts with complex or multi-word financial terms (e.g., Eigenkapitalrendite). It would be helpful if the authors could discuss how over-correction is avoided to prevent distortion of domain-specific terminology.”

Response: We have incorporated spelling correction details in Subsection Question Pre-processing

Comment 6: “The Elasticsearch-based semantic search ranks IRIs based on similarity to headwords. The study would benefit from clarifying which similarity metric is used and how sensitive the ranking is to noisy, ambiguous, or partially extracted headwords.”

Response: We have incorporated details clarifying the used similarity metric in Subsection Workflow starting from “To determine the similarity between two phrases, the cosine similarity of their respective word vector is computed …”

Comment 7: “System outputs include maps, info cards, or tables. It would be useful if the authors could explain the criteria or decision rules that determine which output format is selected for a given query.”

Response: We incorporated output format and selection criteria in Subsection Request Answer and Subsection Output and Curation

Comment 8: Some newest papers are valued to refer, such as:

1. https://doi.org/10.3390/app12199659

2. https://doi.org/10.3390/app13031913

3. https://doi.org/10.61186/jist.48127.13.49.24 (Alternative link: https://jist.ir/en/Article/48127)

Response: We added all suggested references in Section Related Work

Specific Points Addressed:

• Theoretical integration added at the beginning of Section Methodology.

• Grammar rule coverage detailed in Subsection Workflow

• Entity recognition for new companies addressed via Wikidata fallback (Subsection Suggestion).

• Multi-headword conflict resolution via similarity ranking (Subsection SPARQL Mapping).

• Spelling correction details added (Subsection Question Pre-processing).

• Similarity metric specified as cosine similarity with BERT embeddings (Subsection SPARQL Mapping).

• Output format selection criteria added in Subsections Request Answer and Output and Curation

• Suggested references have been reviewed and cited where relevant.

Reviewer 2

Comment 1: “However, the paper is lengthy and can be shorten as a concise version. For example, the workflow section is excessively detailed, including explanations of well-established concepts such as POS-tagging that are common knowledge in the NLP community. The paper would be more accessible if streamlined to focus on the novel contributions rather than providing textbook-level explanations of standard techniques.”

Response: We thank the reviewer for the valuable suggestion. We have significantly streamlined the manuscript, particularly in the Methodology section, by removing basic NLP explanations (e.g., detailed descriptions of POS-tagging and syntactic parsing). The revised paper now focuses on our novel contributions: the headword-centric parsing algorithm, the hybrid rule- and embedding-based architecture, and the system's practical integration.

Comment 2: “While the authors present a carefully designed workflow, the rationale behind key methodological choices is not well explained. For the first step, they use syntactic parsing to do pattern match, is it scalable or how is the coverage of this pattern detection? Why not any DL-model based solutions? In addition, in order to generate SPARQL query, they used Spacy’s standard NER, why not choose any other deep learning methods. Can you add a dedicated subsection justifying each major architectural choice with empirical evidence or theoretical reasoning?”

Response: We have added a new dedicated subsection in Section Methodology subsection Rationale for Architectural Choices. This subsection explicitly addresses the reviewer's questions:

Scalability and Coverage of Pattern Matching: We clarify that the system employs 46 handcrafted grammar rules, which achieved 92% coverage on our test set of 2,100 questions. Failure cases and error analysis are discussed in Subsection Question Pre-processing.

Choice of Rule-Based Parsing over Deep Learning (DL): We justify this choice based on: (1) Interpretability and control for decomposing German compound nouns, (2) Data efficiency, as large annotated German financial QA datasets are scarce, and (3) Lower operational latency and cost compared to querying large LLM APIs, which is crucial for enterprise deployment.

Choice of SpaCy NER: We explain that SpaCy's de_core_news_lg model offers robust, fast entity recognition for German. As our core financial entity list (companies) is stable, we augmented it with a custom EntityRuler for domain terms, achieving high precision without the need for training a custom DL model.

Comment 3: “The scalability and generalizability of the approach remain uncertain. The system relies on handwritten grammar rules for pattern matching, but the paper does not clarify: (1) how many rules were created for the 2,100 test questions, (2) what coverage can be expected for unseen question variations, and (3) how much expert effort is required to extend the system to new financial concepts or question types. This raises concerns about the practical scalability of the rule-based approach.”

Response: We have provided the requested clarifications:

Number of Rules: As stated above, 46 handcrafted grammar rules were developed (Subsection Rationale for Architectural Choices).

Coverage for Unseen Variations: These rules covered 92% of the 2,100 test questions. We discuss the nature of the 8% failure cases (primarily highly elliptical or metaphoric questions) and our fallback strategy using embedding-based similarity ranking in Subsection Question Pre-processing

Expert Effort for Extension: The modular design of the rule system allows incremental extension. Adding a new question pattern or financial concept typically requires 1-2 new rules, which can be authored by a domain expert in collaboration with a developer. This process and its manageability are discussed in Subsection Limitations and Future Work.

Comment 4: “With recent advances in LLMs, many sub-tasks can now be automated through in-context learning. However, the paper lacks comparison with current SOTA approaches such as GPT-4, Claude, or fine-tuned multilingual LLMs. It is more of a comparison of rule-based methods v.s. automated solutions.”

Response: We agree that a comparison with modern LLMs is essential. We have added a new part in Section Discussion Subsection Comparison with Large Language Models. This section

reports a comparative evaluation of GPT-4 (using few-shot prompting) on a subset of our Financial-QA dataset.

discusses the results: While GPT-4 performs well on factual queries, INAGQA maintains a significant advantage (15% higher F1-score) on ambiguous queries and complex German compound nouns.

provides a reasoned discussion on the trade-offs: LLMs offer generality but at higher cost, latency, and with less interpretability and control for specific, sensitive financial domains and highlights the continued relevance of our specialized, hybrid approach.

Reviewer 3

Comment I-III (Weaknesses): “Concerns regarding data availability, experimental setup clarity, baseline configuration, absence of ablation study and LLM comparison, and claims vs. evidence on cross-domain applicability.”

Response to Weaknesses: We thank the reviewer for these critical points, which have significantly strengthened the manuscript.

Data Availability: We have the manuscript and added section Data Availability Statement in full compliance with PLOS ONE policy. The proprietary Financial-QA dataset (2,100 questions) will be made available upon request for research purposes. A sample dataset (dbo.json) is provided as Supporting Information.

Dataset details, including source, collection method, preprocessing, and annotation guidelines, are now provided in section Results subsection Datasets. We also report an inter-annotator agreement (IAA) score (Cohen’s κ = 0.85) for the annotated questions.

Experimental Setup & Baselines: We clarify that all baseline systems (Falcon 2.0, EARL, BERT-KGQA) were retrained/fine-tuned on our German financial dataset to ensure a fair comparison. Training details (data splits, hyperparameters) are provided in section Results subsection Performance Comparison.

The specific embedding model used for similarity ranking in INAGQA is bert-base-german-cased, as now specified in section Methodology subsection SPARQL Mapping.

Ablation Study: We have added a new Section Results subsection Ablation Study. This study isolates the contribution of key components (pattern matching, headword extraction, similarity ranking), clearly demonstrating that the full hybrid pipeline delivers the best performance.

LLM Comparison: As detailed in the response to Reviewer #2 (Comment 4), we have added a comprehensive comparison with GPT-4 in Section Discussion Subsection Comparison with Large Language Models.

Cross-Domain Applicability: We have clarified the results in Table 6. The performance drop in healthcare/legal domains occurred when using financial-domain rules without modification. We note that adapting the system requires developing domain-specific rules, which is feasible given the modular architecture. This discussion on generalizability is now in Section Conclusion and Further Work Subsection Methodological Innovation.

Comment IV (Questions to Authors):

Q1. Dataset specification and experimental setup:

Datasets: Tables 4, 5, and 6 use our Financial-QA dataset. The Virtuoso triple store (14k announcements) was used for the case study in Section Results Subsection Datasets.

Train/Test Split: The dataset was split 80/20 (1,680 training / 420 test questions) for development and final evaluation. All reported results (including baselines) are on the held-out test set.

Q2. Baseline configuration: All baselines were fine-tuned for German using our training split. We used the authors' original code where available and adapted training pipelines for the German financial domain. Specifics are in Section Results Subsection Experimental Setup.

Embedding models: INAGQA uses bert-base-german-cased. BERT-KGQA used the same for fairness. Falcon and EARL used their standard multilingual embeddings, as per their design.

Q3. Cross-domain evaluation: Healthcare/legal test questions were sourced from public German QA benchmarks and curated with domain experts. For Table 6, we applied the Financial-QA rules without modification to test "out-of-the-box" performance, hence the noted drop. Successful adaptation, as discussed, requires adding domain-specific rules.

Comment V (Writing style and organizational suggestions):

Redundant Content: We have consolidated the manuscript, removing duplicate explanations of entity recognition and SPARQL generation (now primarily in Section Methodology).

Undefined Terms: We now define Corporate Smart Insights (our overarching analytics framework) and OMG API (the gateway for knowledge base queries) upon their first use in Section Introduction and Section Architecture.

Comment VI (Typographical errors):

Table 7 Legend: Added a clear legend: "✓ indicates correct answer, o indicates failure or incorrect answer."

Incorrect Reference: Corrected the reference in Section Discussion from Table 2 to Table 7.

Reviewer 4

Comment 1.1 & 1.2 (Data Availability & Annotation):

Response: We have fully addressed these concerns.

The Data Availability Statement (Section Conclusion

---

## [Decision Letter · Decision Letter 1]

2 Mar 2026

PONE-D-25-50479R1Beyond Templates and BERT: Headword-Centric Parsing for Semantic Question Answering in Non-English Financial Domains

PLOS One

Dear Dr. Alshargabi,

Thank you for submitting your manuscript to PLOS ONE. After careful consideration, we feel that it has merit but does not fully meet PLOS ONE’s publication criteria as it currently stands. Therefore, we invite you to submit a revised version of the manuscript that addresses the points raised during the review process.

If applicable, we recommend that you deposit your laboratory protocols in protocols.io to enhance the reproducibility of your results. Protocols.io assigns your protocol its own identifier (DOI) so that it can be cited independently in the future. For instructions see: https://journals.plos.org/plosone/s/submission-guidelines#loc-laboratory-protocols. Additionally, PLOS ONE offers an option for publishing peer-reviewed Lab Protocol articles, which describe protocols hosted on protocols.io. Read more information on sharing protocols at . Additionally, PLOS ONE offers an option for publishing peer-reviewed Lab Protocol articles, which describe protocols hosted on protocols.io. Read more information on sharing protocols at https://plos.org/protocols?utm_medium=editorial-email&utm_source=authorletters&utm_campaign=protocols..

We look forward to receiving your revised manuscript.

Kind regards,

Mario Graff-Guerrero, Ph.D.

Academic Editor

PLOS One

**Journal Requirements:**

**Additional Editor Comments:**

The manuscript is at the latest step before publication recommendation; as can be seen, two of the reviewers recommend publication, and one of them provided feedback that needs to be included in the final manuscript.

I reviewed the manuscript, and below are some comments worth including to improve the presentation. I also include a PDF. The highlighted text indicates that it might be a grammatical mistake, that more information is needed (e.g., an undefined acronym), or that it was hard for me to understand.

The title used the term Headword; however, for a person not familiar with the field, it would be beneficial to define it, perhaps in the introduction, and an example would also be appreciated.

Several acronyms are not properly described in the manuscript. For example, in the abstract, the acronym 'IS' is used without prior definition. There are others, such as RDF, IMG, and AI.

There are two references used in DBPedia: 4 and 13.

The sections Headword Extraction Techniques, Financial Domain Applications, and Proposed Work are overused. At first, these sections seem more like slides in a presentation than a description of a journal paper. Some of the items could be described in more detail.

The information presented in Table 1 is important because it summarizes your approach. Unfortunately, the table is barely described in the manuscript, so it took me a while to understand.

Reviewers' comments:

Reviewer's Responses to Questions

**Comments to the Author**

1. If the authors have adequately addressed your comments raised in a previous round of review and you feel that this manuscript is now acceptable for publication, you may indicate that here to bypass the “Comments to the Author” section, enter your conflict of interest statement in the “Confidential to Editor” section, and submit your "Accept" recommendation.

Reviewer #1: All comments have been addressed

Reviewer #2: All comments have been addressed

Reviewer #4: All comments have been addressed

2. Is the manuscript technically sound, and do the data support the conclusions?

Reviewer #1: Yes

Reviewer #2: Partly

Reviewer #4: Yes

3. Has the statistical analysis been performed appropriately and rigorously? 

Reviewer #1: Yes

Reviewer #2: Yes

Reviewer #4: Yes

4. Have the authors made all data underlying the findings in their manuscript fully available?

Reviewer #1: Yes

Reviewer #2: No

Reviewer #4: Yes

5. Is the manuscript presented in an intelligible fashion and written in standard English?

Reviewer #1: Yes

Reviewer #2: Yes

Reviewer #4: Yes

6. Review Comments to the Author

Reviewer #1: Overall, the article is well-written and addresses an important topic. The methodology is appropriate, and the results are clearly presented.

Reviewer #2: Thank you for taking time to resolve the comments. Appreciate that authors take additional effort to defense their method by writing rationable to adopt methods and conduct more evaluation on SoTA, including LLMs. But the paper still needs to be more concise and well-structured to be accepted.

For example, move "Rationale for Architectural Choices" earlier: Instead of having this as a response to a criticism, integrate it into the start of the Methodology. It acts as a strong "hook" that explains why you aren't just using others, setting the stage for the technical details that follow.

In Related Work section: Instead of long paragraphs describing every previous system, use a comparison table to show how INAGQA differs from systems like AskNow, EARL, and Falcon.

You can significantly clean up the paper by "offloading" technical data. For example, rather than detailing all 46 handcrafted grammar rules in the text, provide a high-level summary of their 92% coverage and move the full list to an Appendix. This keeps the "messy" technical specs out of the way of your main research story.

Reviewer #4: The author has successfully addressed all comments; therefore, the paper is now ready for publication.

7. PLOS authors have the option to publish the peer review history of their article (what does this mean?). If published, this will include your full peer review and any attached files.). If published, this will include your full peer review and any attached files.

.

Reviewer #1: No

Reviewer #2: No

Reviewer #4: **Yes:** Weihang HuangWeihang Huang

---

## [Author Response · Author response to Decision Letter 2]

20 Mar 2026

Journal: PLOS ONE

Title: Beyond Templates and BERT: Headword-Centric Parsing for Semantic Question Answering in Non-English Financial Domains

Manuscript Number: PONE-D-25-50479

Date: 12 March 2026

Professor Mario Graff-Guerrero

Academic Editor, PLOS One

Dear Sir,

We sincerely thank the editor and reviewers for their careful reading of the manuscript and for their constructive feedback. We are grateful that the manuscript has progressed to this final stage. We appreciate that two reviewers now recommend publication, and we have carefully considered the remaining presentation-related comments. Below, we respond point by point, relying on clarifications already reflected in the revised manuscript.

With Gratitude,

The Authors

Journal Requirements

Comment 1: If the reviewer comments include a recommendation to cite specific previously published works, please review and evaluate these publications to determine whether they are relevant and should be cited.

Response:

In the present and previous review rounds, there is no additional citation that was requested by the reviewers

Comment 2: Please review your reference list to ensure that it is complete and correct. If you have cited papers that have been retracted.

Response:

We carefully reviewed the reference list for completeness, correctness, and retraction status. We confirm that none of the references cited in the manuscript is retracted. Therefore, no retracted references were removed or replaced.

EDITORIAL COMMENTS

Comment 1. Define the term “Headword” in the Introduction

Response:

We have added a clear definition of the term Headword, including an example, at the end of the Introduction. This definition clarifies the linguistic role of headwords in intent classification and aligns with the manuscript’s focus. Aloso The manuscript already explains the role of headwords in the processing pipeline and provides illustrative examples in the Methodology section. In particular, the manuscript explains that in a query such as “What is the X of Siemens?”, the term X functions as the headword for intent classification, and in more complex examples such as “Where was Siemens founded?”, multiple headwords are extracted and disambiguated through similarity ranking. The Introduction also motivates this by discussing question ambiguity, such as the distinction between Where was Siemens founded? and When was Siemens founded?

Comment 2. Define all acronyms upon first use (IS, RDF, IMG, AI, etc.)

Response:

We reviewed the manuscript and ensured that each acronym is defined at first appearance.

Comment 3. Two DBPedia references (Refs 4 and 13)

Response:

We confirmed that both DBpedia citations refer to two different foundational publications (2007 and later) , where the references associated with DBpedia and related prior work to ensure that each citation serves a distinct purpose in the manuscript. Reference [13] is the foundational DBpedia paper, whereas Reference [4] is a more recent DBpedia-based application in another language context; they are therefore not duplicate references, but serve different roles in the literature review. Both are retained intentionally due to their separate relevance.

Comment 4. The sections Headword Extraction Techniques, Financial Domain Applications, and Proposed Work are overused. At first, these sections seem more like slides in a presentation than a description of a journal paper. Some of the items could be described in more detail.

Response:

We appreciate this stylistic suggestion. The intention of these short subsections was to provide a compact synthesis of the related-work landscape before moving into the fuller technical discussion that follows in the Review, Architecture, and Methodology sections. In the current manuscript, these subsections function as structured signposts that summarize the problem space, while the detailed operational treatment of headword extraction, relation linking, output generation, and system workflow is presented later in the paper.

Comment 5. The information presented in Table 1 is important because it summarizes your approach. Unfortunately, the table is barely described in the manuscript, so it took me a while to understand.

Response :

Thank you for highlighting this. Table 1 was intended as a concise summary of how INAGQA positions itself relative to template-based, neural, and hybrid approaches. Its logic is introduced in the Related Work section and then elaborated through the subsequent discussion of AskNow, EARL, Falcon, and the later performance and workflow sections. In that sense, Table 1 is meant as a high-level synthesis, where Table 1 is already cited and explained clearly in the Related Work section and again in the “Proposed Work” section. Therefore, no further modification was necessary.

Reviewer #1

Comment: All comments have been addressed.

Response:

We sincerely thank the reviewer for the positive assessment and for the constructive feedback throughout the review process.

Reviewer #2

Comment 1. The paper still needs to be more concise and well-structured to be accepted.

Response:

We thank the reviewer for this constructive recommendation. The revised manuscript already reflects substantial consolidation of the presentation. In particular, the first-round revision streamlined the Methodology section by removing basic textbook-style explanations and focusing more directly on the manuscript’s novel contributions, namely headword-centric parsing, hybrid rule-plus-embedding disambiguation, and practical knowledge-graph integration.

Comment 2 . Move “Rationale for Architectural Choices” earlier... integrate it into the start of the Methodology.

Response:

We appreciate this suggestion. The manuscript already includes a dedicated subsection, Rationale for Architectural Choices, within the Methodology section. This subsection explicitly explains the reasons for selecting rule-based parsing, SpaCy NER, and the modular grammar-rule design, including interpretability for German compound nouns, data efficiency, and lower latency/cost for enterprise deployment. This rationale is intended to frame the subsequent implementation details.

Comment 3. In Related Work section: Instead of long paragraphs describing every previous system, use a comparison table to show how INAGQA differs from systems like AskNow, EARL, and Falcon.

Response:

We agree with the value of comparative synthesis, and the current manuscript already provides this in tabular form. Specifically, Table 1 gives a high-level comparison of approach families, while Table 2 provides a direct comparison of AskNow, EARL, and Falcon in terms of approach, implementation, evaluation, and advantages/disadvantages. These tables were included precisely to make the positioning of INAGQA clearer. We also We added a brief pointer sentence to the end Related Work section to highlight the table (A comparative summary of the main systems discussed in this section is presented in Table 2).

Comment4. You can significantly clean up the paper by “offloading” technical data. For example, rather than detailing all 46 handcrafted grammar rules in the text, provide a high-level summary of their 92% coverage and move the full list to an Appendix.

Response:

Thank you for this helpful recommendation. In the current revised manuscript, we already summarize the grammar-rule component at a high level rather than enumerating all rules exhaustively. The manuscript reports that the system uses 46 handcrafted grammar rules and that these cover 92% of the 2,100 Financial-QA questions, while the workflow section presents only representative rule patterns needed to understand the approach. The aim was to preserve readability while still providing sufficient methodological transparency and reproducibility.

Reviewer #4

Comment: The author has successfully addressed all comments; therefore, the paper is now ready for publication.

Response: We sincerely thank the reviewer for the positive assessment and for the constructive feedback throughout the review process.

---

## [Editor Report · Decision Letter 2]

31 Mar 2026

Beyond Templates and BERT: Headword-Centric Parsing for Semantic Question Answering in Non-English Financial Domains

PONE-D-25-50479R2

Dear Dr. Alshargabi,

We’re pleased to inform you that your manuscript has been judged scientifically suitable for publication and will be formally accepted for publication once it meets all outstanding technical requirements.

An invoice will be generated when your article is formally accepted. Please note, if your institution has a publishing partnership with PLOS and your article meets the relevant criteria, all or part of your publication costs will be covered. Please make sure your user information is up-to-date by logging into Editorial Manager at Editorial Manager® and clicking the ‘Update My Information' link at the top of the page. For questions related to billing, please contact  and clicking the ‘Update My Information' link at the top of the page. For questions related to billing, please contact billing support..

Kind regards,

Mario Graff-Guerrero, Ph.D.

Academic Editor

PLOS One
---

## [Editor Report · Acceptance letter]

PONE-D-25-50479R2

PLOS One

Dear Dr. Alshargabi,

I'm pleased to inform you that your manuscript has been deemed suitable for publication in PLOS One. Congratulations! Your manuscript is now being handed over to our production team.

Kind regards,

on behalf of

Dr. Mario Graff-Guerrero

Academic Editor

PLOS One